



# mesas.py v1.0: A flexible Python package for modeling solute transport and transit times using StorAge Selection functions

Ciaran J. Harman[1,2] and Esther Xu Fei[1]

[1]Department of Environmental Health and Engineering, Johns Hopkins University Baltimore, MD, USA
[2]Department of Earth and Planetary Sciences, Johns Hopkins University Baltimore, MD, USA
**Correspondence:** Ciaran J. Harman (charman1@jhu.edu)

**Abstract.** StorAge Selection transport theory has recently emerged as a framework for representing material transport through a control volume. It can be seen as a generalization of transit time theories and lumped parameter models to allow for arbitrary time-variability of the rate of material flow in and out of the control volume, and in the transport dynamics. SAS is currently the state-of-the-art approach to interpreting tracer transport. Here we present `mesas.py` , a Python package implementing the

SAS framework. `mesas.py` allows SAS functions to be specified using several built-in common distributions, as a piecewise-linear CDF, or as a weighted sum of any number of such distributions. The distribution parameters and weights used to combine them can be allowed to vary in time, allowing SAS functions of arbitrary complexity to be specified. `mesas.py` simulates tracer transport using a novel mass tracking scheme and can account for first order reactions and fractionation. We present a number of analytical solutions to the governing equations and use these to validate the code. For a benchmark problem the

timestep-averaging approach of the `mesas.py` implementation provides a 15x reduction in mass balance errors compared to a previous implementation of SAS.

## 1   Introduction

StorAge Selection (SAS) is a theoretical framework for modeling transport dynamics through spatially integrated systems (control volumes). It is applicable in any system where it is reasonable to assume that the bulk material flowing out of a

system (at rate $Q(t)$) at some time $t$ is some conservative mixture of the bulk material that flowed in at earlier times (at rate $J(t)$). For example, it is often reasonable to assume that the streamflow and evapotranspiration leaving a watershed are some conservative mixture of precipitation that fell on that watershed at earlier times. The conservative bulk material in that case is simply the water comprising the rainfall, streamflow, and evapotranspiration. The development of this theory and related issues in watershed hydrology have been recently reviewed in Benettin et al. (2022).

SAS is a generalization of the idea of a transit time distribution (TTD), which have proved useful in a wide range of disciplines including chemical engineering (Ross et al., 2006), transportation engineering (Tyworth and Zeng, 1998), groundwater hydrology (Dupas et al., 2020; Rinaldo et al., 2015; Danesh-Yazdi et al., 2018), surface water hydrology (Stockinger et al., 2016; Rodriguez and Klaus, 2019; Rodriguez et al., 2021), medicine (Rossum et al., 1989), and others. However TTD have previously required that the bulk material flow through the system be approximately steady (i.e. $J(t) = Q(t) =$ a constant). SAS





relaxes this assumption in a rigorous and general way, so that it can (in principle) be used to characterize transport through any system where the bulk material flow is conserved. However to date SAS functions have not been widely adopted in practice. In part this is due to the perception that they are too complex and data-hungry.

Our objective here is to provide detailed documentation of `mesas.py`, a Python implementation of SAS functions that is easy to use, highly flexible, sophisticated, and computationally accurate. This implementation is already the basis of online

teaching resources (Harman, 2020), and we hope to develop more in the future. It is essential therefore that there exist a peer-reviewed publication supporting and documenting the software.

In a typical forward-modeling use-case, we wish to predict the concentration $C_Q(t)$ of a conservative tracer in the bulk material outflow $Q(t)$, which is assumed to be a conservative mixture of previous bulk material inflows $J(t)$ in which the tracer concentration was $C_J(t)$. If so, the outflow concentration $C_Q(t)$ will be some weighted average of past values of $C_J(t)$.

The transit time distribution $p_Q(T,t)$ gives those weights:

$$C_Q(t) = \int\limits_0^\infty C_J(t-T) p_Q(T,t) dT \qquad (1)$$

SAS provides a means to calculate the time varying distribution $p_Q(T,t)$ for a given system. An overview of SAS and related approaches can be found in Botter (2012); Harman (2015); Rinaldo et al. (2015); Benettin and Bertuzzo (2018).

The basic equations required to calculate $p_Q(T,t)$ (discussed in Section 2 below) are not especially difficult to solve nu-

merically, but some care is required. An implmentation of SAS in MATLAB (`tran-SAS`) is already available for MATLAB (Benettin and Bertuzzo, 2018). `mesas.py` replicates the functionality of `tran-SAS`, but offers the following features:

- `mesas.py` offers an extremely flexible framework for specifying SAS functions, allowing them to be arbitrarily complex and time-varying. This includes the ability to specify SAS functions as a time-varying weighted sum of other functions (Rodriguez and Klaus, 2019) and as a (time-varying) piecewise-linear CDF with any number of segments

- `mesas.py` uses a novel mass-tracking approach that estimates solute/tracer storage and outflow rates as part of the solution, not through a subsequent convolution integral

- `mesas.py` estimates the timestep-averaged transit times and mass fluxes using a Runge-Kutta 4th order method, and provides superior numerical accuracy and mass balance accounting (as we shall demonstrate)

- `mesas.py` allows for time-varying first-order reactions and time-varying solute/tracer fractionation

- `mesas.py` is implemented in Python and Fortran and is designed to be easy to install (through `conda-forge`) and user-friendly

The governing equations of the SAS framework are given in Section 2 of this paper, including the novel approach to solute/tracer mass tracking. Calculating the storage and release of solutes/tracers continuously in tandem with calculation of the TTD (rather than using the convolution after the TTD has been obtained) makes incorporating reactions and fractionation into





SAS function simple and intuitive. Section 3 gives details of the code, including the numerical implementation, the method for specifying SAS functions, and precedures for running the code.

In Section 4 of the paper we test the code against a number of benchmarks in the form of analytical solutions to the governing equations. These include cases of steady and unsteady flow. We compare the accuracy of `mesas.py` against that of `tran-SAS` for the unsteady flow case.

## 2  Governing equations

To estimate $p_Q$ and solve equation (1) two key pieces of information are required: 1) timeseries of inflows $J(t)$ and outflows $Q_q(t)$ (there may be more than one outflow, hence the subscript index $q$), and 2) *SAS functions* $\Omega_q$ (one for each outflow $q$) that capture the way each outflow is drawn from the water of different ages available to be removed from storage. The inflow and outflow data are used to solve expressions of conservation of mass that describe how the age distribution of the material in

storage changes over time as material is added and removed. The SAS functions are needed to calculate this solution because they characterize the relative rate that material of different relative ages is selected for removal.

### 2.1  Conservation Laws

#### 2.1.1  Conservation law for the bulk material flows

The bulk flow is the material that makes up the inflows and outflows from the system, carrying tracers and other species of

material with it. Typically in hydrologic applications the bulk flow is water. As is typical in hydrology we assume the water is incompressible so we can refer to units of volume for convenience, but the framework is valid for any conservative bulk flow as long as fluxes, storages and concentrations are expressed in consistent compatible units.

The conservation equation for the bulk flow can be obtained by considering an incremental volume $s_T(T,t)$ that has an age $T$ at time $t$. It therefore entered at time $t_i = t - T$. Note that $s_T(T,t)$ has units of volume (or mass) *per time*, as it is the amount

that entered in an infinitesimal increment of time. If the inflow rate is $J(t)$ at some time in the past $t = t_i$, then $s_T(0,t) = J(t)$. Over time, the quantity of bulk flow residing in storage represented by $s_T$ depletes due to outflows. Assuming that outflows are indexed by $q$, and each has a outflow rate $Q_q$ and transit time distribution $p_q(T,t)$, then the time evolution of $s_T$ from some initial time $t_i$ to the present time $t$ is given by:

$$\frac{d}{dt}s_T(t-t_i,t) = J(t)\delta(t-t_i) - \sum_q Q_q(t)p_q(t-t_i,t) \tag{2}$$

where $\delta(\cdot)$ is the Dirac delta distribution. Typically the derivative on the left here is broken up into two terms, like so:

$$\frac{d}{dt}s_T(t-t_i,t) = \frac{\partial s_T}{\partial t} + \frac{\partial s_T}{\partial T} \tag{3}$$





However the form given in (2) serves to remind us that these two derivatives can be thought of as representing the rate that $s_T$ changes as it simultaneously moves through time and ages. We can think of it moving along a characteristic curve that is a straight line in age-time space, with a slope of 1 unit of age per unit of time and passing through the point $(T, t) = (0, t_i)$. The

computational method of `mesas.py` is based on numerically integrating along this characteristic curve.

Integrating $s_T$ over all ages up to some age $T$ gives the cumulative form $S_T$, known as the *age-ranked storage*:

$$S_T(T, t) = \int\limits_0^T s_T(\tau, t)\, d\tau \tag{4}$$

This is the volume of bulk material residing in storage that is younger than $T$ at time $t$. The bulk material conservation equation is often expressed in terms of this cumulative quantity and $P_q(T, t)$, the cumulative form of $p_q(T, t)$. Equation (2) can

be obtained from the cumulative form by taking the derivative with respect to $T$.

$S_T$ is also essential for solving (2) through its role in evaluating the SAS function (see Section 2.2 below). Therefore even though the primary state variable `mesas.py` solves for is $s_T(T, t)$, the code must also keep track of the accumulating values of $S_T(T, t)$.

### 2.1.2 Conservation law for solutes

Consider a conservative solute or tracer that travels ideally with the bulk material. We can define $m_T(T, t)$ as the incremental tracer mass that entered at time $t_i$ and is now remaining in storage ($m_T$). We can also define a notion of 'age-ranked concentration' in storage as the increment of age-ranked solute mass per increment of age-ranked storage:

$$C_T(T, t) = \frac{m_T(T, t)}{s_T(T, t)} \tag{5}$$

Note that this is not a concentration in the usual sense. It may not correspond to an actual measurable concentration anywhere

in the system if material of different ages have intermingled sufficiently. However it does (by definition) equal the input concentration $C_J(t)$ just as water enters the system, thus:

$$m_T(0, t) = C_T(0, t) s_T(0, t) = C_J(t) J(t) \tag{6}$$

It also gives the effective concentration of the solute in the increment of water at age $T = t - t_i$ that is contributing to each outflow, and thus controls the mass flux out for a given age increment, which we will term $\dot{m}_q$:

$$\dot{m}_q(T, t) = Q_q(t) C_T(T, t) p_q(T, t) \tag{7}$$





Putting these together, we can write a conservation law for the solute as:

$$\frac{d}{dt}m_T(t-t_i,t) = J(t)C_J(t)\delta(t-t_i) - \sum_q \dot{m}_q(t-t_i,t) \tag{8}$$

This equation is analogous to (2), but instead of tracking the bulk material along a characteristic curve in time-age space, it tracks the mass of solute/tracer.

### 2.1.3 Accounting for fractionation and reactions

We can easily generalize (8) in two ways.

First, we can account for the effect of fractionation in the outflows. Sometimes the concentration of a solute in an outflow is different from its concentration in storage. For example, chloride (which has been used as a tracer in catchment studies (Harman, 2015)) can leave in discharge, but cannot leave with evapotranspiration. Thus the effective concentration of chloride in the evaporative flux must be zero. A less extreme example is where stable water isotope ratios in evaporation tend to be lighter than those in the water left behind.

We can account for this fractionation in a simple way by assuming the concentration of the solute in outflow $q$ is some (possibly time-varying) multiple $\alpha_q(t)$ of the concentration in storage. To accomplish this we modify equation (7) to include this:

$$\dot{m}_q(T,t) = \alpha_q(t)Q_q(t)C_T(T,t)p_q(T,t) \tag{9}$$

When $\alpha_q = 1$ there is no fractionation. $\alpha_q < 1$ will result in reduced concentrations in the given outflow, and $\alpha_q = 0$ excludes the solute from the outflow. It is also possible to set $\alpha_q > 1$ if the solute is preferentially entrained in the outflow.

Second, we can account for first order reactions. The change in $C_T$ resulting from mass introduced or removed by such a reaction can be modeled as:

$$\frac{dC_T}{dt} = k_1(C_{eq} - C_T) \tag{10}$$

From this we can define a reaction term $\dot{m}_R$ as:

$$\dot{m}_R(T,t) = s_T(T,t)\frac{dC_T}{dt} = k_1(t)(C_{eq}(t)s_T(T,t) - m_T(T,t)) \tag{11}$$

where $k_1$ is a first-order reaction rate, and $C_{eq}$ is an equilibrium concentration (both of which may vary in time).

Including fractionation and the reaction terms in the solute conservation law gives:

$$\frac{d}{dt}m_T(t-t_i,t) = J(t)C_J(t)\delta(t-t_i) - \sum_q \dot{m}_q(T,t) + \dot{m}_R(T,t) \tag{12}$$



where $\dot{m}_q$ is given by (9) and $\dot{m}_R$ is given by (11)

The actual outflow concentration at time $t$ is obtained by integrating $\dot{m}_q(T,t)$ over all ages $T \leq T_{max}$ (where $T_{max} = t$ usually, but may be less than $t$), plus the 'old water' flux:

$$C_q(t) = \frac{1}{Q_q(t)} \int_0^{T_{max}} \dot{m}_q(T,t)dT + C_{old} \times (1 - P_q(T_{max},t)) \tag{13}$$

where $C_{old}$ is the concentration assigned to all the water in storage whose age is greater than $T_{max}$.

## 2.2  StorAge Selection (SAS) functions

The equations above cannot be solved on their own, as there are fewer equations than unknowns. The SAS functions provide the required additional relationship linking the age-ranked storage and the transit time distribution.

Given a volume of age-ranked storage $S_T$ representing all the bulk material in a control volume with an age of $T$ or less, the
cumulative SAS function $\Omega_q$ is defined as a function that gives the fraction of outflow $Q_q$ drawn from $S_T(T,t)$. This is also (by definition) the fraction of discharge whose with an age of $T$ or less, which is simply $P_q(T,t)$. Thus we can write:

$$P_q(T,t) = \Omega_q(S_T,t) \tag{14}$$

where $S_T = S_T(T,t)$. That is, the SAS function and the cumulative transit time distribution both give the fraction of discharge with age $T$ or less, but the SAS function expresses the age in terms of age ranked storage $S_T(T,t)$, rather than age $T$.
This has proved to be very useful since $P_q(T,t)$ varies in time due to variations in fluxes, $\Omega_q(S_T,t)$ only varies when the manner in which storage turns over varies. In many applications SAS functions have been approximated by a variety of continuous distributions with good results (Benettin et al., 2022). We will discuss several that have been implemented in `mesas.py` in the next section.

By taking the derivative of the equation above and applying the chain rule, we can see that:

$$p_q(T,t)\delta T = \omega_q(S_T,t)\delta S_T \tag{15}$$

where $\omega_q$ is the density form of $\Omega_q$. The left hand side of this equation is the rate of discharge of water with ages between $T$ and $T + \delta T$. The right hand side is the rate water is removed from the age-ranked storage between $S_T$ and $S_T + \delta S_T$. These are related by:

$$\delta S_T = \delta T \frac{\partial S_T}{\partial T} = \delta T s_T \tag{16}$$





If our age step is $\delta T$ the relationships above allow us to determine the transit time distribution as:

$$p_q(T,t) = \omega_q(S_T,t)s_T(T,t) \tag{17}$$

and therefore solve the conservation law (2).

### 2.2.1   Continuous SAS functions available in `mesas.py`

The SAS function must be specified so that it accurately captures how a system turns over, releasing storage as bulk outflow.

At present, three continuous distributions commonly used for specifying SAS functions are available built-in to `mesas.py` : the Beta, Kumaraswamy, and Gamma distributions. More details on each distribution are given below.

These distributions each have at least two parameters: a *location* parameter $S_{\min}$ and a *scale* parameter $S_0$. These parameters both have units of storage, and serve to shift and scale the values of $S_T$ into a normalized form:

$$x = \frac{S_T - S_{\min}}{S_0} \tag{18}$$

with $x = 0$ if $S_T < S_{\min}$. In `mesas.py` the values of $S_0$, $S_{\min}$ and all other parameters can be given as constant values, or different values for every timestep can be provided.

– **Beta distribution:**The CDF of the Beta distribution is given by:

$$\text{Beta}(S_T; S_{\min}, S_0, \alpha, \beta) = \frac{B(x; \alpha, \beta)}{B(\alpha, \beta)} \quad S_{\min} \leq S_T \leq S_{\min} + S_0 \tag{19}$$

where $B(x; \alpha, \beta)$ and $B(\alpha, \beta)$ are the incomplete and complete Beta functions, and $\alpha, \beta > 0$. This distribution has been

used to represent the SAS function of systems whose total active storage volume $S(t)$ has been estimated (Benettin et al., 2022). In those cases $S_m$ is set to zero and $S_0 = S(t)$

– **Kumaraswamy distribution:** The CDF of the Kumaraswamy distribution is given by:

$$\text{Kumaraswamy}(S_T; S_{\min}, S_0, a, b) = 1 - (1 - x^a)^b \quad S_{\min} \leq S_T \leq S_{\min} + S_0 \tag{20}$$

where $a, b > 0$. This distribution has also been used to represent the SAS function of systems whose total active storage

volume has been estimated.

– **Gamma distribution:** The Gamma distribution is given by:

$$\text{Gamma}(S_T; S_{\min}, S_0, \alpha) = \frac{\Gamma(\alpha, x)}{\Gamma(\alpha)} \quad \text{for} \quad S_{\min} \leq S_T < \infty \tag{21}$$





This distribution has been used to represent the SAS function of systems whose total volume is unknown, and the right tail of the SAS function is assumed to taper exponentially for sufficiently large $S_T$ (Harman, 2015; Benettin et al., 2022).

In addition, any distribution specified by the `scipy.stats` library can be used in `mesas.py` by converting its CDF into a piecewise-linear form (see next section). This is done automatically within the `mesas.py` . The accuracy of the results obtained this way may be poor.

### 2.2.2   Uniform and piecewise-linear SAS functions in `mesas.py`

SAS functions can also be specified as a piecewise-linear CDF with $N$ segments. These linear segments join control points
$(S_{T\,q,0}, \Omega_{q,0})$, $(S_{T\,q,1}, \Omega_{q,1})\ldots(S_{T\,q,N}, \Omega_{q,N})$. To ensure the result is a probability distribution, we require $0 = \Omega_{q,0} \leq \Omega_{q,1} \leq \ldots \leq \Omega_{q,N} = 1$, and $0 \leq S_{T\,q,0} \leq S_{T\,q,1} \leq \ldots \leq S_{T\,q,N}$. The SAS function is specified by giving these control points, which may vary in time. The PDF $\omega_q(S_T, t)$ is piecewise constant:

$$
\omega_q(S_T, t) = \begin{cases}
\dfrac{\Omega_{q,1} - \Omega_{q,0}}{S_{T\,q,1} - S_{T\,q,0}} & S_{T\,q,0} \leq S_T < S_{T\,q,1} \\[2em]
\dfrac{\Omega_{q,2} - \Omega_{q,1}}{S_{T\,q,2} - S_{T\,q,1}} & S_{T\,q,1} \leq S_T < S_{T\,q,2} \\[2em]
\ldots \\[2em]
\dfrac{\Omega_{q,N} - \Omega_{q,N-1}}{S_{T\,q,N} - S_{T\,q,N-1}} & S_{T\,q,N-1} \leq S_T < S_{T\,q,N} \\[2em]
0 & \text{otherwise}
\end{cases}
\tag{22}
$$

The parameters of the $N$-segment piecewise SAS function are the $N+1$ values of $S_{T\,q,n}$ and $N-1$ values of $\Omega_{q,n}$ (recall
that $\Omega_{q,0} = 0$ and $\Omega_{q,N} = 1$).

### 2.2.3   SAS functions built from weighted sums of components

`mesas.py` also allows a SAS function to be specified as a (time-varying) weighted sum of component SAS functions specified in any of the available ways. This approach was first suggested by Rodriguez and Klaus (2019), and Wilusz et al. (2020) provided evidence supporting its validity.





Given $M$ *component* SAS functions (indexed by $m$) defined using one of the methods presented above, the overall SAS function can be obtained as:

$$\omega_q(S_T, t) = \sum_{m=1}^{M} f_{q,m}(t)\omega_{q,m}(S_T, t) \tag{23}$$

where $f_{q,m}(t)$ is a (possibly time-varying) weight. These weights must be provided as inputs to the model. The weights should sum to 1 at each timestep (though this is not enforced by `mesas.py`).

## 3    Methods

### 3.1    Numerical implementation

The numerical implementation of the governing equations in `mesas.py` is reminiscent of a numerical finite-volume scheme. We will assume that timesteps $\Delta t$ and agesteps $\Delta T$ are equal.

First, agestep-averaged forms of the state variables are obtained by integrating in $T$ over an interval of past input times 205 $[t_i - \Delta T, t_i]$:

$$s_i(t) = \frac{1}{\Delta T} \int_{t_i - \Delta T}^{t_i} s_T(t - \tau, t)d\tau \tag{24}$$

$$m_i(t) = \frac{1}{\Delta T} \int_{t_i - \Delta T}^{t_i} m_T(t - \tau, t)d\tau \tag{25}$$

For notational consistency we define $S_i(t) = S_T(t - t_i, t)$. We can use these to write agestep-averaged forms of (2) and (12):

$$\Delta T \frac{ds_i}{dt} = J(t) \int_{t_i - \Delta T}^{t_i} \delta(t - \tau)d\tau - \sum_q Q_q(t) \int_{t_i - \Delta T}^{t_i} p_q(t - \tau, t)d\tau \tag{26}$$

$$\Delta T \frac{dm_i}{dt} = J(t)C_J(t) \int_{t_i - \Delta T}^{t_i} \delta(t - \tau)d\tau - \sum_q \left( \int_{t_i - \Delta T}^{t_i} \dot{m}_q(t - \tau, t)d\tau \right) - \int_{t_i - \Delta T}^{t_i} \dot{m}_R(t - \tau, t)d\tau \tag{27}$$

There are a few things to unpack here. First, note that due to the properties of the Dirac $\delta$-function the integral $\int_{t_i - \Delta T}^{t_i} \delta(t - \tau)d\tau$ is 1 if $t_i - \Delta T < t < t_i$, and is zero otherwise. Thus for all times after $t_i$ the first term disappears in both equations above. This behavior can be represented using the indicator function $\mathbf{1}_{[t_i - \Delta T, t_i)}(t)$, which we will write as $\mathbf{1}_i(t)$ for shorthand.





The agestep-averaged TTD $p_{qi}(t)$ can be expressed in terms of the cumulative TTD, and thus in terms of the SAS function

as:

$$
\begin{aligned}
p_{qi}(t)\Delta T = \int_{t_i-\Delta T}^{t_i} p_q(t-\tau,t)d\tau &= P_q(t-t_i+\Delta T,t) - P_q(t-t_i,t) \\
&= \Omega_q(S_T(t-(t_i-\Delta T),t),t) - \Omega_q(S_T(t-t_i,t),t) \\
&= \Omega_q(S_i(t)+s_i(t),t) - \Omega_q(S_i(t),t)
\end{aligned}
\tag{28}
$$

The timestep-averaged values of $\dot{m}_q$ and $\dot{m}_R$ can then be obtained if we are willing to approximate the age-ranked concentration $C_T$ is constant over the interval $\Delta T$ at $C_T = m_i/s_i$. This amounts to approximating the concentration $C_J$ in the corresponding input bulk material as constant over this interval.

Thus we can express the governing equations as the set of ODEs:

$$
\Delta T \frac{ds_i}{dt} = J(t)\mathbf{1}_i(t) - \sum_q Q_q(t)p_{qi}(t)\Delta T
\tag{29}
$$

$$
= f_s(s_i, S_i, t)\Delta T
\tag{30}
$$

$$
\Delta T \frac{dm_i}{dt} = J(t)C_J(t)\mathbf{1}_i(t) - \frac{m_i(t)}{s_i(t)}\left(\sum_q \alpha_q(t)Q_q(t)p_{qi}(t)\Delta T\right) - k_1(t)(C_{eq}(t)s_i(t) - m_i(t))
\tag{31}
$$

$$
= f_m(m_i, s_i, S_i, t)\Delta T
$$

If the right hand side of these equations were functions of the state variables $s_i$ and $m_i$ and a number of time-variable

coefficients ($J$, $Q_q$, $C_J$, $\alpha_q$, $k_1$, $C_{eq}$) then these would be ODEs. The dependence of the SAS function on the cumulative value $S_i(t)$ complicates matters only slightly.

Numerical solution of these equations involves two core tasks:

- Estimating the rates of change $f_s(s_i, S_i, t)$ and $f_m(m_i, s_i, S_i, t)$

- Using these to estimate the state variables $s_i$ and $m_i$ (and $S_i$) at a future time

Let us assume $t = j\Delta t$ and $T = i\Delta T$, and recall that age and time steps are equal in size. Now define $\mathbf{s}_i^j = s_i(j\Delta t)$, and define $\mathbf{m}_i^j$ similarly. Particular care must be taken to ensure that an accurate value of $S_T$ is used to evaluate the SAS function. Let

$$
\mathbf{s}_i^j = \sum_{k=i+1}^j \mathbf{s}_k^j \Delta t
\tag{32}
$$





Note that by this definition the value of $\mathbf{S}_i^j$ depends on $\mathbf{s}_{i+1}^j$ but not on $\mathbf{s}_i^j$. For the RK4 method the state variables must be evaluated at an intermediate point half way between the regular time steps. The value of $\mathbf{S}_i^{j+\frac{1}{2}}$ can be estimated as:

$$\mathbf{S}_i^{j+\frac{1}{2}} = \frac{\mathbf{S}_i^j + \mathbf{S}_i^{j+1}}{2} \tag{33}$$

But note that $\mathbf{S}_i^{j+1}$ can be found only after an estimate of $\mathbf{s}_i^{j+1}$ is obtained. This suggests that in order to calculate how $\mathbf{s}_i^j$ becomes $\mathbf{s}_i^{j+1}$ we must first have determined both $\mathbf{s}_{i+1}^j$ and $\mathbf{s}_{i+1}^{j+1}$. In other words, for an accurate numerical solution we need to know the fate of younger material before we can determine the fate of older material. Because of this, `mesas.py` solves $j \to j+1$ for all $i$ before moving on to $i+1$, and so on.

`mesas.py` uses a Runge-Kutta 4th-order method (RK4) to estimates the value of $\mathbf{s}_i^{j+1}$ as follows:

$$
\begin{aligned}
k_1 &= f_s(\mathbf{s}_i^j, \mathbf{S}_i^j, j\Delta t) \\
k_2 &= f_s(\mathbf{s}_i^j + \tfrac{1}{2}k_1\Delta t, \mathbf{S}_i^{j+\frac{1}{2}}, (j+\tfrac{1}{2})\Delta t) \\
k_3 &= f_s(\mathbf{s}_i^j + \tfrac{1}{2}k_2\Delta t, \mathbf{S}_i^{j+\frac{1}{2}}, (j+\tfrac{1}{2})\Delta t) \\
k_4 &= f_s(\mathbf{s}_i^j + k_3\Delta t, \mathbf{S}_i^{j+1}, (j+1)\Delta t) \\
k_* &= \frac{k_1 + 2k_2 + 2k_2 + k_1}{6} \\
\mathbf{s}_i^{j+1} &= \mathbf{s}_i^j + k_*\Delta t
\end{aligned}
\tag{34}
$$

Similar steps are followed to estimate $\mathbf{m}_i^{j+1}$ using $f_m(m_i, s_i, S_i, t)$. During the calculation the intermediate values of $p_q$, $\dot{m}_q$ and $\dot{m}_R$ are also tracked and timestep-averaged according to the same scheme. The state variables are held in memory in arrays whose columns are times, and whose rows are ages. Thus stepping through timesteps $j$ for a fixed input time $i$ corresponds with stepping along a diagonal of a state variable matrix.

`mesas.py` allows initial conditions to be specified for the age ranked storage $s_T$ and solute mass $m_T$. This is useful for restarting calculations or spinning up the simulation. The initial conditions are supplied as a vector of values $\mathbf{s}_k^0$ and $\mathbf{m}_k^0$ for ages $k$. Each entry represents the value of $s_T$ and $m_T$ at $t = 0$ averaged over each age interval, and is used to populate the first column of the matrices holding the state variables.

`mesas.py` proceeds by solving all the timesteps $j \to j+1$ for the first age step before moving on to second, and so forth. Typically, when there are $N$ time steps the solution is found for $N$ age steps also. This is sometimes excessive since the contributions of bulk flow from the start of the simulation to the final timestep may be negligible. The number of age steps can also be larger than $N$, though this will only have value if an initial condition with values of $\mathbf{s}_i^0$ for more than $> N$ ages. Note that all outflow whose age is unknown is assigned concentration $C_{old}$.

## 3.2 Model specification and input structure

Inputs to `mesas.py` come in two main forms:





- parameters specifying the SAS function(s), solute properties, and other model settings

- timeseries of inflows, outflows, and other variables

The parameters are specified using a nested data structure that can be stored and read from a JSON-formatted text file, or fed into a model object instance directly as a Python dictionary. The timeseries can be provided as a .csv text file, or as a Pandas dataframe.

The parameter data structure consists of a dictionary of key:value pairs, where a 'key' is an immutable label (typically a string), and a 'value' is an object that can be retrieved from the dictionary using the associated key. The values can themselves
be dictionaries, allowing for a nested structure to the data.

The top level dictionary in the parameter specification must have a key `"sas_specs"`. The associated value must be a dictionary of SAS specifications. It may also have two optional entries: `"solute_parameters"` provides information about solutes to be routed through the model, and `"options"` can be used to set a number of model options.

### 3.2.1   SAS function specification

A basic example of the `"sas_specs"` key:value pair is shown below:

```
"sas_specs":{
"Q":{
"Q SAS function 1":{
"func": "gamma",
"args": {
"loc": 0.0,
"scale": "S_scale",
"a": 0.6856
}
},
"Q SAS function 2":{
"func": "beta",
"args": {
"loc": 0.0,
"scale": 150.0,
"a": 1.0,
"b": 3.0
}
}
},
"ET":{
"ET SAS function":{
"ST": [0.0, "S_ET"],
"P":  [0.0, 1.0]
}
```



```
}
},
```

The `sas_spec` dictionary should contain one key for each bulk flux out of the control volume, and each key must exactly match the heading of a column in the timeseries dataset giving that flux rate. In the example above, `mesas.py` would expect the timeseries dataset to contain columns `Q` and `ET`.

Each of the keys naming a bulk flux in `sas_specs` is associated with a dictionary specifying the SAS functions for that flux. That dictionary can also include multiple SAS functions, which are combined together using time-varying weights. In the example above `mesas.py` would expect to find columns in the timeseries dataset titled `"Q SAS function 1"` and `"Q SAS function 2"` containing weights to multiply each SAS function. These weights should add up to 1, though this is not checked. If this dictionary contains only one key:value pair then it is not necessary to provide a weights column in the timeseries dataset.

Presently, each SAS function can be specified in three different ways:

- As a Gamma, Beta, or Kumaraswamy distribution

- Using any distribution from `scipy.stats`

- As a piecewise linear cumulative distribution function (CDF)

The Gamma, Beta, or Kumaraswamy distributions are coded into the core computational code, while `scipy.stats` distributions will be approximated as piecewise linear CDFs. In either case the distribution is selected based on the value associated with `"func"`. In the example above gamma and beta distributions are combined to produce the SAS function for outflow `Q`.

The distribution parameters are given by the dictionary associated with the key `"args"`. The expected contents of this varies between distributions (see table 1). Any parameter value can be specified as a fixed number, or can be allowed to vary in time. Time-varying parameters are given as a string identical to a column in the timeseries dataset where the time varying values are provided. In the example `sas_specs` above the scale parameter of the Gamma distribution used in `Q SAS function 1` is set to `S_scale`. This tells `mesas.py` to use the timeseries of values found in that column of the input dataset for the scale parameter.

To use a distribution from `scipy.stats` the key:value pair `"use":"scipy.stats"` should be included. An optional parameter `"nsegments"` sets the number of segments used to approximate the distribution. Note that this approach is included for convenience, but is not recommended when the tails of the SAS distribution are important for the problem being considered, as they may not be well captured by the piecewise linear CDF.

Alternatively, the SAS function can be specified as a piecewise linear CDF. In the example above, this option is used to specify a uniform SAS function for ET using a single linear segment. The cumulative age-ranked storage values `"ST"` and corresponding cumulative probabilities `"P"` (varying from 0 to 1) must be provided as lists of increasing values. Any of the values in these lists may be allowed to vary in time by instead providing a string corresponding to the heading of a column in the input timeseries dataset – see for example `"S_ET"` in the `"ET SAS function"` above.





**Table 1.** Relationship between the parameters in equations (19), (20), (21) and the keys used to specify the value of these parameters in the SAS function specification.

| Gamma distribution (21) | Beta distribution (19) | Kumaraswamy distribution (20) |
|---|---|---|
| $S_{min} \leftrightarrow$ `"loc"` | $S_{min} \leftrightarrow$ `"loc"` | $S_{min} \leftrightarrow$ `"loc"` |
| $S_0 \leftrightarrow$ `"scale"` | $S_0 \leftrightarrow$ `"scale"` | $S_0 \leftrightarrow$ `"scale"` |
| $\alpha \leftrightarrow$ `"a"` | $\alpha \leftrightarrow$ `"a"` | $a \leftrightarrow$ `"a"` |
| | $\beta \leftrightarrow$ `"b"` | $b \leftrightarrow$ `"b"` |

### 3.2.2 Solute parameters

Solutes properties are given in a dictionary associated with the top-level key `"solute_parameters"`. The keys in this dictionary should correspond with columns in the timeseries dataset giving inflow concentrations. Each key should be associated with a dictionary giving additional parameters. If defaults are to be used, the associated dictionary may be empty, and simply given as `{}`. An example is given below:

```
"solute_parameters":{

    "Cl mg/l":{
        "C_old": 7.11,
        "alpha": {"Q": 1.0, "ET": 0.0}
    }
},
```

In this case `mesas.py` will look for a timeseries of solute inflows in column `"Cl mg/l"` and produce predictions of the outflow concentrations associated with this input. Two additional parameters are specified. `"C_old"` gives the old water concentration $C_{old}$ and `"alpha"` corresponds to the $\alpha_q$ partitioning parameter in (9). In the given example no chloride can leave the system through ET, as the corresponding value of $\alpha$ is zero. See Table 2 for more information.

### 3.2.3 Options

Additional options can also be set in the `"options"` dictionary of the parameter inputs. The available options are described in Table 3.

### 3.2.4 Timeseries

The timeseries input can be provided as a .csv file, or as a Python Pandas dataframe. The order of the columns is not important but the column names should be consistent with references to timeseries data in the SAS function specification, solute parameters, and options.





**Table 2.** Description of the keys that may optionally be in the dictionary associated with each solute in the `"solute_parameters"` parameter input dictionary.

| Key | Symbol(s) | Description |
|---|---|---|
| `"C_old"` | $C_{old}$ | Old water concentration. This will be the concentration of all water released of unknown age. If `sT_init` is not specified, this will be all water in storage at $t = 0$. If `sT_init` is specified, it will be all water older than the last non-zero entry in `sT_init`. Default is $0.0$. Cannot be set as time-varying. |
| `"k1"` | $k_1$ | First-order reaction rate constant in equation (11). Default is $0.0$. May be time-varying if a time-series column name is provided. |
| `"C_eq"` | $C_{eq}$ | Equilibrium concentration in equation (11). Default is $0.0$. May be time-varying if a timeseries column name is provided. |
| `"alpha"` | $\alpha_q$ | A dictionary giving partitioning coefficients for each outflow as in (9) . Default is $1.0$. Dictionary keys must correspond to named outflow columns in the SAS specification. Each $\alpha_q$ may be time-varying if a timeseries column name is provided. |
| `"sT_init"` | $s_T(T,0)$ | List or array of values specifying the initial age-ranked storage distribution in the system. This is useful if the system is initialized by some sort of spin-up. Each entry is age-ranked storage in an age interval of duration $\Delta t$. |
| `"mT_init"` | $m_T(T,0)$ | List or array of values specifying the initial age-ranked mass in the system. This is useful if the system is initialized by some sort of spin-up. Each entry is age-ranked mass in an age interval of duration $\Delta t$. If `mT_init` is specified, `sT_init` must also be specified in the `options`, and be of the same length. The element-wise ratio `mT_init/sT_init` gives the age-ranked concentration $C_T$ of the water in storage at time zero. Note that if `sT_init` is specified but `mT_init` is not, the concentrations associated with each non-zero value of `sT_init` will be zero. Default is `None`. Cannot be set as time-varying. |





**Table 3.** Description of the keys that may optionally be in the dictionary associated with each solute in the `"solute_parameters"` parameter input dictionary.

| Key | Description |
|---|---|
| `"influx"` | String, default is "J". Gives the name of the column in the timeseries dataset containing the inflow rate. |
| `"dt"` | Timestep $\Delta t$, such that $\Delta t$ multiplied by any of the fluxes in the timeseries dataset gives the total volume of flux over the timestep. Default is 1.0. Cannot be set as time-varying. |
| `"n_substeps"` | Integer, default is 1. Number of substeps used in each timestep of the calculation. Subdividing the timesteps can increase the numerical accuracy of the solution and address some numerical issues, at the cost of longer run times. Note that the substep calculations are not retained in the output – only aggregate timestep results are provided. |
| `"max_age"` | Integer, default is the length of the timeseries dataset. The maximum number of age steps that will be calculated. This controls the number of rows in the output matricies. Set to a smaller value than the default to reduce calculation time (at the cost of replacing calculated concentrations of older water with the value of `"C_old"`) |
| `"sT_init"` | List or array, default is zero array of the length of the timeseries dataset. Initial distribution of age-ranked storage (in density form). Useful for starting a run using output from another model run, e.g. for spin up. If the length of this array is less than the length of the timeseries dataset, then `"max_age"` will be set to the length of `"sT_init"`. |
| `"verbose"` | Boolean, default is `false`. Print information about the calculation progress. |
| `"warning"` | Boolean, default is `true`. Print warnings about calculation issues. |
| `"debug"` | Boolean, default is `false`. Print a very large amount of information about the calculation progress. Do not use. |



For example, to be consistent with the specifications given in the example in Sections 3.2.1 and 3.2.2, the input dataframe would have the following columns:

- `"Q"` and `"ET"`: outflow rates (e.g. discharge and evapotranspiration) at each timestep. These are assumed to be average rates over the timestep (rather than instantaneous rates at the start or end)

- `"J"`: Average inflow rate over each timestep

- `"Cl mg/l"`: Inflow concentration at each timestep

- `"Q SAS function 1"`, `"Q SAS function 2"`: weights associated with the two component SAS functions that
will be combined to give the SAS function for `"Q"`. Note that a column for `"ET SAS function"` is not required since there is only one component

- `"S_ET"` and `"S_scale"`: time varying parameters of the SAS functions

After running, the output timeseries would include the following new columns:

- `"Cl mg/l -> Q"`

- `"Cl mg/l -> ET"`

Representing the concentration of the solute in those outflow fluxes. The values of `"Cl mg/l -> ET"` would all be zero, since the partitioning coefficient `"alpha"` associated with that solute and outflow was set to zero.

### 3.3 Running the model and querying results

The model is setup and run by instantiating a model object provided with all the needed input data, then calling its `run` method:

```
1  from mesas.sas.model import Model
2  my_model = Model(data_df='/path/to/data.csv', config='/path/to/config.json')
3  my_model.run()
```

The timeseries inputs and outputs will then be available in a dataframe accessible as an attribute of the model object. For example, this will allow you to plot the input and output concentrations:

```
import matplotlib.pyplot as plt
# Extract the timeseries
C_in = my_model.data_df['Cl mg/l']
C_out = my_model.data_df['Cl mg/l --> Q']
6  t = my_model.data_df.index
# Make the plots
```





```
plt.plot(t, C_in, label = "Cl in precip")
plt.plot(t, C_out, label = "Cl in discharge")
# Finishing touches
plt.legend(frameon=False)
plt.xlabel('time')
plt.ylabel('Cl [mg/l]')
```

Users can access further results using accessor functions. These can return the values for a particular time-step, age-step, or input time. The latter is useful for examining how water that entered at a particular time evolves in time. If none of these are given, the entire array is returned. Both density (sT, pQ, mT, mQ, mR) and cumulative (ST, PQ, MT, MQ, MR) forms are available.

```
# Make an array of ages to plot against
T = my_model.options['dt'] * np.arange(my_model.options['max_age'])
# Extract and plot the TTD at a particular timestep
pQ = my_model.get_pQ(timestep=100, flux='Q')
plt.figure()
plt.step(T, pQ, where='post')
# Extract and plot the volume of water in storage with an age less than 90
ST = my_model.get_ST(agestep=90)
plt.figure()
plt.step(T+1, ST, where='pre')
# Extract and plot the concentration of water in storage as it evolves due to evapoconcentration
sT = my_model.get_sT(inputtime=328)
mT = my_model.get_mT(inputtime=328, sol='Cl mg/l')
CT = mT/sT
plt.step(t, CT, where='post')
```

More information on these functions is available in the documentation.

## 4  Code validation and comparison

To validate the numerical implementation `mesas.py` was tested against several analytical benchmark solutions. Six of these are analytical solutions for different SAS functions under steady flow. Additional benchmark solutions for unsteady flow are identical to ones presented for `tran-SAS` in Benettin and Bertuzzo (2018), and can therefore be used for comparison.

### 4.1  Validation against benchmarks: steady flow

#### 4.1.1  Approach

For certain SAS functions it is possible to find a closed-form expression for the corresponding TTD under steady flow. For the six cases considered here the details of the derivations are given in Appendix A, and the mathematical results are listed in Table



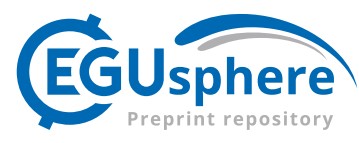

| Case | Flux $Q(t) =$ | SAS function $\Omega(S_T, t) =$ | Continuous TTD $p_Q(T, t) =$ | Discrete TTD $pq_{i,j} = \frac{1}{\Delta t\,\delta_j} \times$ |
|---|---|---|---|---|
| Uniform | $Q$ | Uniform$(0, S_0)$ | $\dfrac{Q}{S_0} e^{-T\frac{Q}{S_0}}$ | $\begin{cases} \delta + e^{-\delta} - 1 & i = 0 \\ e^{-(1+i)\delta}\left(e^{\delta} - 1\right)^2 & i > 0 \end{cases}$ |
| Exponential | $Q$ | Gamma$(1, S_0)$ | $\dfrac{Q}{S_0}\left(T\dfrac{Q}{S_0} + 1\right)^{-2}$ | $\begin{cases} \delta + \log\left(\dfrac{1}{\delta+1}\right) & i = 0 \\ \log\left(\dfrac{(\delta i+1)^2}{(\delta(i-1)+1)(\delta(i+1)+1)}\right) & i > 0 \end{cases}$ |
| Biased old | $Q$ | Kumaraswamy$(2,1)$ or Beta$(2,1)$ on $S_T \in [0, S_0]$ | $\dfrac{2Q\tanh\left(T\frac{Q}{S_0}\right)}{S_0\cosh^2\left(T\frac{Q}{S_0}\right)}$ | $\begin{cases} \delta - \tanh(\delta) & i = 0 \\ \dfrac{2\sinh^2(\delta)\tanh(\delta i)}{\cosh((i-1)\delta)\cosh((i+1)\delta)} & i > 0 \end{cases}$ |
| Biased young | $Q$ | Kumaraswamy$(1,2)$ or Beta$(1,2)$ on $S_T \in [0, S_0]$ | $2\dfrac{Q}{S_0}\left(T\dfrac{Q}{S_0} + 1\right)^{-3}$ | $\begin{cases} \dfrac{\delta^2}{\delta+1} & i = 0 \\ \dfrac{2\delta^2}{(\delta(i-1)+1)(\delta i+1)(\delta(i+1)+1)} & i > 0 \end{cases}$ |
| Partial bypass | $Q$ | Kumaraswamy$(\frac{1}{2},1)$ or Beta$(\frac{1}{2},1)$ on $S_T \in [0, S_0]$ | $-\dfrac{Q}{2S_0}\left(W\left(-e^{-\frac{QT}{2S_0}-1}\right)^{-1} + 1\right)^{-1}$ | $\begin{cases} \delta + M(1) - 1 & i = 0 \\ M(i-1) - 2M(i) + M(i+1) & i > 0 \end{cases}$ $M(\ell) = W\left(-e^{-\frac{\delta\ell}{2}-1}\right)\left(W\left(-e^{-\frac{\delta\ell}{2}-1}\right) + 2\right)$ |
| Partial piston | $Q$ | Kumaraswamy$(1,\frac{1}{2})$ or Beta$(1,\frac{1}{2})$ on $S_T \in [0, S_0]$ | $\dfrac{Q}{2S_0}$ on $T \in [0, 2S_0/Q]$ | $\begin{cases} \dfrac{\delta^2}{4} & i = 0 \\ \dfrac{\delta^2}{2} & i > 0 \end{cases}$ |
| Uniform & Time-varying flux | $Q(t)$ | Uniform$(0, S(t))$ | $\dfrac{J(t-T)}{S(t)}\exp\left(-\int_{t-T}^{t}\dfrac{Q(\tau)}{S(\tau)}d\tau\right)$ | $\begin{cases} e^{-\delta_j}\phi_j + \delta_j - 1 & i = 0 \\ \dfrac{S_{j-i}}{S_j}\exp\left(-\sum_{k=0}^{i}\delta_{j-k}\phi_{j-k}\right)\left(e^{(\delta_{j-i}+\eta_{j-i})\phi_{j-i}} - 1\right) \times (e^{\delta_j\phi_j} - 1) & i > 0 \end{cases}$ |

**Table 4.** Analytic solutions for the continuous and discrete TTD for a number of benchmark SAS functions with Uniform, Gamma, and Beta distributions. The discrete form is obtained by averaging the value of $p_Q$ over each age step and time step. In the time-variable cases it is assumed that fluxes are constant over each timestep, so $S_{j+1} = S_j + \Delta t(J_j - Q_j)$. For notational convenience we have defined $\delta_j = \Delta_T Q_j/S_j$, $\kappa = k\Delta t$, $\eta_{ji} = S_{j+1}/S_j - 1$ and $\phi_j = \log(\eta_{ji}+1)/\eta_{ji}$ if $\eta_{ji} \neq 0$, otherwise $\phi_j = 1$.



4. Several of these have been found previously Botter (2012); Harman (2015); Berghuijs and Kirchner (2017), though others
are new.

The six cases (also shown in the top row of Figure 1) are a uniform distribution, an exponential distribution, a 'biased old'
and a 'biased young' distribution which encode a bias for older and younger storage that varies linearly with age rank in
storage, and a 'partial piston' and a 'partial bypass' distribution (both of which encode a strong preference for the oldest and
youngest storage respectively). The latter four scenarios are special cases of both Beta and Kumaraswamy distributions.

To assess the validity of our implementation of the numerical solution against these closed-form expressions we can either
a) use very fine timesteps and thus more closely approximate the continuous result, or b) find an analytical form of the discrete
solution. The latter is preferable, since we can compare the numerical and analytical results directly, rather than asymptotically
at the limit of small time steps. We have therefore taken the additional step of obtaining discrete versions of each expression
(rightmost column of Table 4), which when convolved with a synthetic timeseries of input concentrations yield the average
output concentration over each timestep. These exact values can be compared directly to the numerical results, which are also
intended to represent the average value over each timestep.

In each scenario the flow rate was set to $J(t) = Q(t) = 1$ and the timestep to $\Delta t = 0.1$. The value of the scale parameter
was set to $S_0 = 5$, and an offset of $S_{min} = 1$ was used. Consequently outflow concentrations are delayed relative to inflows by
5 timesteps. Inflow concentrations were synthetically generated as independent identically distributed random values (white
noise) normally distributed with a mean of 1.0 and a standard deviation of 1.0. Initial concentration in storage ($C_{old}$) was set
to 1.0. The n_substeps parameter was initially set to 1, then increased to 10 to examine how a greater number of numerical
substeps improved the solution accuracy.

### 4.1.2 Results

The second row of Figure 1 (g - l) plots the tracer concentration predicted by mesas.py (blue and orange lines as labeled in
the first row) and analytical benchmark solutions (black dash line). The random inputs are most smoothed by the 'biased old'
case, and retain much of the input variability in the 'partial bypass' case. The last two rows of Figure 1 present the percent
errors relative to the benchmark when one (m - r) or 10 (s - x) numerical steps are taken each timestep.

When n_substeps=1, the root mean square errors for most cases are on the order of $10^{-6}$ for all but the the uniform,
where they were around $10^{-9}$, and the partial bypass case, whose errors are closer to $10^{-3}$. The steeply-varying SAS function
associated with young water in the partial bypass case appears to present a serious numerical challenge.

When n_substeps=10, the errors decrease by a factor of around 100 in the exponential and partial piston case, but the
decrease is smaller for others. Notably the error in the partial bypass case is reduced by a factor of 40, demonstrating that
increasing n_substeps can improve numerical accuracy in this difficult case (at the cost of some computational speed).



**Figure 1.** Results of the benchmark runs under steady flow. (a-f) The SAS functions used in each case. In each case `"loc"=1` and `"scale"=5`. The steady flow rate was $Q = 1$, and the timestep was $\Delta t = 0.1$. (g-l) The predictions produced by `mesas.py` and the analytical benchmark solutions (black dashed lines). The inflow concentrations were Gaussian random variables with a mean of 1 and a standard deviation of 1. The initial concentration in storage was 1. (m-r) Absolute error relative to the benchmark with `"n_substeps"=1`. (s-x) Error with `"n_substeps"=10`. Note that the annotation '1e-6' on an axis indicates that the axis values are multiples of $10^{-6}$.





## 4.2 Validation against benchmarks: unsteady flow

### 4.2.1 Approach


The power of the SAS approach comes from its ability to handle time-variable inflows and outflows. The bottom row of Table 4 gives the analytical solution for the case where the SAS function is uniform but the flow rate is time-variable. The general analytical solution was presented in Botter (2012), but the discrete form given here is novel. The discrete form is derived from the general case by assuming that inflows and outflow are constant over a timestep, so that the storage varies linearly. Further,


the solution given is not the instantaneous pdf $p_Q$, but rather $p_Q$ averaged over an agestep/timestep along a characteristic curve. It therefore gives a precise estimate of the expected value of the fraction of discharge over each timestep drawn from inputs in previous timesteps.

This benchmark was used to validate the `mesas.py` code for the same dataset Benettin and Bertuzzo (2018) used to validate the performance of `tran-SAS`. The dataset was downloaded from the repository cited in Benettin and Bertuzzo


(2018), and includes eight years of 12-hourly precipitation, discharge, and evapotranspiration data. Benettin and Bertuzzo (2018) generated input concentrations by adding noise to a seasonal sinusoidal signal. The evapotranspiration was assumed to be drawn uniformly from the total storage in all simulations. Total storage at the end of each timestep is calculated from the water balance assuming an initial storage $S_{init}$. The $S_0$ parameter used in `mesas.py` was calculated by averaging the total storage at the start and end of each timestep.

### 4.2.2 Results


Figure 2a shows the input solute concentration, and the output concentration predictions of `mesas.py` and `tran-SAS` for the case where the discharge SAS function is uniform and $S_{init} = 1000$mm. The model predictions are visually indistinguishable from one another. Figure 2b also showcases the effect of activating some of the features of `mesas.py` : the ability to account for first-order reactions (for the case where the reaction rate is $3 \times 10^{-4}$/hr) and fractionation (for the case where $\alpha_{ET} = 0.8$,


so that evapotranspiration enriches the tracer concentration in storage).

Closer inspection of the residuals between the model concentration predictions and the analytical benchmarks reveals differences between the performance of `tran-SAS` and `mesas.py` . Figure 3a,b show the timeseries and distribution of errors for `tran-SAS` and `mesas.py` for the case where $S_{init} = 1000$mm. Though the overall distribution of absolute error magnitudes is similar, `tran-SAS` produces relatively large errors about 15% of the time. Overall the root mean square error (RMSE) of


`mesas.py` is 0.4% (of the output standard deviation), while for `tran-SAS` it is 1.6%.

These differences become larger when we consider the error in the solute mass flux, as shown in Figure 3c,d. The RMSE of `mesas.py` is 0.016%, while for `tran-SAS` it is about 15 times larger, at 0.21%.

The differences can be almost entirely attributed to the fact that `tran-SAS` provides estimates of the *instantaneous* transit time distribution at the end of each timestep, while `mesas.py` estiamtes *timestep-averaged* values. It is also possible to obtain


the TTD for the end of the timestep from `mesas.py` output, and use them to estimate outflow concentrations. Those estimates have errors (shown in green in Figure 3d) very similar to those of `tran-SAS`, as we would expect.



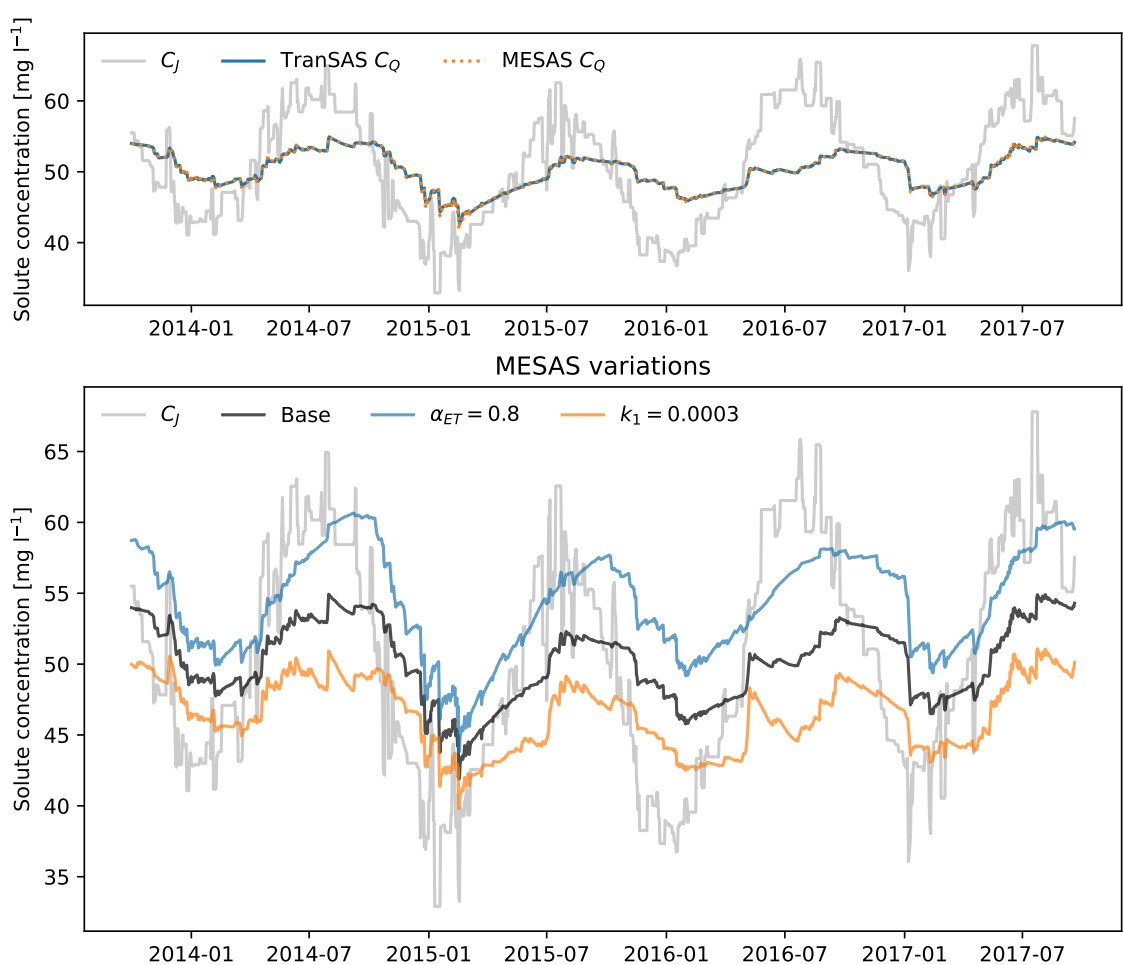

**Figure 2.** `mesas.py` application on `tran-SAS` timeseries. Upper: $C_Q$ estimations from `mesas.py` (orange dash) and `tran-SAS` (blue line) after four-year spinup period; grey line represents input concentration from precipitation ($C_J$) Lower: variations of `mesas.py` simulation. Grey line: input concentration $C_J$ from precipitation; Black line: base case, using the same setting as `tran-SAS` model; Blue line: change $\alpha_{ET} = 0.8$ upon base case; Orange line: add first-order reaction rate $k_1 = 0.0003$ upon base case.





**Figure 3.** Comparison of `mesas.py` with `tran-SAS` for a benchmark problem presented by Benettin and Bertuzzo (2018). a) normalized RMSE of outflow concentration estimates from `mesas.py` (orange) and `tran-SAS` (blue) relative to the timestep-averaged analytical solution with $S_{init} = 1000$ and $k = 1$; b) Absolute concentration error cdf for `mesas.py` and `tran-SAS`; c,d) as with (a,b), but showing errors in the mass flux, rather than the concentration; e) effect of varying $k \in [0.2, 0.3, 0.5, 1, 1.2, 1.5, 2, 3]$ with $S_{init} = 1000$, on the difference between each code's predictions and a run of `mesas.py` with 10 substeps as an high-accuracy estimate; f) with $k = 1$, effect of varing $S_{init} \in [300, 500, 1000, 2000]$, on the concentration estimates of `tran-SAS`, `mesas.py` relative to to the analytical solution.

`mesas.py` performs better than `tran-SAS` for other configurations of the problem, though the size of the difference changes. The normalized root-mean-squared-error (RMSE) of each implementation are shown in Figure 3f for four different





values of initial storage. The results show that normalized RMSE is larger for both codes when storage is small, however
`mesas.py` has a lower RMSE than `tran-SAS` in all cases.

We can also compare the performance of these models for the case where the discharge SAS function is not uniform. Again following Benettin and Bertuzzo (2018) we consider the case where $\Omega_Q(S_T, t) = \left[ \frac{S_T(T,t)}{S(t)} \right]^k$. This is equivalent to a Kumaraswami distribution with $S_{\min} = 0$, $S_0 = S(t)$, $a = k$ and $b = 1$. The required parameter JSON file is given below:

```
"sas_specs":{
"Q":{
"Q SAS function":{
"func": "kumaraswamy",
"args": {
"loc": 0.0,
"scale": "S",
"a": "k",
"b": 1.0
}
}
},
"ET":{
"ET SAS function":{
"ST": [0.0, "S"],
"P":  [0.0, 1.0]
}
}
},
"solute_parameters":{
"C":{
"C_old": 50,
}
},
"options":{
"influx": "J"
"n_substeps": 1
},
```

The timeseries dataset includes columns `Q`, `J`, `ET`, `S`, `k`, and input concentrations `C`.
Since analytical solutions are unavailable for this more general case the results obtained from `tran-SAS` and `mesas.py` were compared against a higher-accuracy `mesas.py` solution (obtained by setting `n_substeps` to 10). The RMSE for a range of values of $k$ are shown in Figure 3c. `mesas.py` RMSE was consistently lower than `tran-SAS`, though errors in both cases were higher for small $k$.





## 5   Conclusion

SAS transport theory provides a very general framework for modeling material transport through control volumes (Benettin et al., 2022). At its core, it is based on a statement of conservation of mass of bulk material of different ages. This must be augmented with a SAS function that captures how outflows preferentially remove bulk material from storage according to the rank of its age. The `mesas.py` code presented here implements this theory, allows SAS functions to be expressed in a very flexible way, and solves the underlying equations with high accuracy with regard to mass balance.

The code is also intended to be user-friendly. A number of resources are available for people, including a free online course is available through HydroLearn. The course, entitled *JHU 570.412 Tracers and transit times in time-variable hydrologic systems: A gentle introduction to the StorAge Selection (SAS) approach* can be found at https://edx.hydrolearn.org/courses/course-v1: JHU+570.412+Sp2020/course/ (free registration required). This course includes three sections of introductory SAS function theory accompanied by a `mesas.py` walk-through.

Further work is needed to augment the code with additional useful tools. Three sets of tools are particularly important. First, tools for generating ensembles of input concentration data. In hydrology, observations of input concentrations are often bulk samples that represent amount-weighted averages over multiple timesteps. These must be disaggregated to the resolution at which we want to run the model. Second, tools for parameterizing SAS functions and fitting them to data, preferably in a way that can adapt to any specification of the SAS function. Third, tools for assessing uncertainty in both the disaggregated inputs, the SAS function shape, and the model predictions.

*Code availability.* `mesas.py` v1.0 is open source and distributed under the terms of the MIT License. The code is available on GitHub here: https://github.com/charman2/mesas. Version 1.0 is tagged as v1.20221012, and is archived at https://doi.org/10.5281/zenodo.7144731. The code is open source, and users are encouraged to use GitHub's issue tracking framework for submitting bug reports and feature requests.

The most up-to-date version of `mesas.py` along with its dependencies can be installed from the command line using `conda` with:

The model can also be installed by building from the source code using `python setup.py install`. A FORTRAN compiler is required to do so (but is not required when installing through `conda`).

Documentation for `mesas.py` is available at https://mesas.readthedocs.io/en/latest/. This documentation is also stored in the GitHub repository.

*Data availability.* No new data is presented in this paper.

## Appendix A: Analytical benchmarks

### A1   Steady-state solutions

At steady state with one outflow and $J = Q$ (2) can be written as:



$$\frac{ds_T}{dT} = -Qp_Q(T) \tag{A1}$$

$$s_T(0) = Q \tag{A2}$$

This is a separable differential equation, which we can integrate twice to give:

$$S_T(T) = Q\left(T - \int_0^T P_Q(\tau)\, d\tau\right) \tag{A3}$$

This gives the steady-state cumulative age-ranked storage in terms of the cumulative transit time distribution $P_Q$.

Alternatively we can use the fact that $p_Q = \omega_Q s_T$ and $s_T = dS_T/dT$ to write the conservation law as:

$$\frac{dS_T}{dT} = Q - Q\Omega_Q(S_T(T)) \tag{A4}$$

$$S_T(0) = 0 \tag{A5}$$

Which can likewise be integrated to obtain the age of water at a given location in age-ranked storage in terms of the SAS function:

$$T = \frac{1}{Q} \int_0^{S_T(T)} \frac{1}{1 - \Omega_Q(\sigma)}\, d\sigma \tag{A6}$$

Using these equations we can (in principle) find the steady-state transit time distributions if we know the SAS function, and

vice versa. In many cases a closed form solution is not possible.

For example, for a uniform distribution $U(0, S_0)$ SAS function $\Omega_Q(S_T) = S_T/S_0$ for $S_T \in [0, S_0]$. Substituting this into (A6) gives $QT = -S_0 \log(1 - S_T/S_0)$, and rearranging gives $S_T(T) = S_0\left(1 - e^{\frac{Q}{S_0}T}\right)$. This is the cumulative age-rank storage when $\Omega_Q$ is a uniform distribution and the flow is steady. Substituting this into the definition of the uniform $\Omega_Q$, yields the cumulative transit time distribution (since $P_Q(T,t) = \Omega_Q(S_T,t)$ by definition). Consequently, we can say that at steady state:

$$\Omega_Q(S_T) = U(0, S_0) \quad \Leftrightarrow \quad P_Q(T) = 1 - e^{-\frac{Q}{S_0}T} \tag{A7}$$

That is, a uniform SAS function is equivalent to an exponential transit time distribution.

A similar set of steps can be used to show that an exponential SAS function (which is a special case of a Gamma distribution with shape parameter $\alpha = 1$) is equivalent to TTD following a Lomax distribution with exponent 1:

$$\Omega_Q(S_T) = \Gamma(1, S_0) \quad \Leftrightarrow \quad P_Q(T) = 1 - (1 + QT/S_0)^{-1} \tag{A8}$$



We were not able to obtain a solution for the general case of a Gamma distribution. Similarly, solutions for the TTD when the SAS function is given by a Beta distribution $B(\alpha,\beta)$ over $0 \leq S_T/S_0 \leq 1$ could only be found for particular values of $(\alpha,\beta)$. For example, with $\alpha = 1$ and $\beta = 2$ we have the "biased young" case:

$$\Omega_Q(S_T) = B(1,2) \quad \Leftrightarrow \quad P_Q(T) = 1 - (1 + TQ/S_0)^{-2} \tag{A9}$$

Which is a Lomax distribution with exponent 2. Similarly the "biased old", "partial bypass", and "partial piston" cases are:

$$\Omega_Q(S_T) = B(2,1) \quad \Leftrightarrow \quad P_Q(T) = \tanh^2\left(TQ/S_0\right) \tag{A10}$$

$$\Omega_Q(S_T) = B(\tfrac{1}{2},1) \quad \Leftrightarrow \quad P_Q(T) = W\left(-e^{-\frac{QT}{2S_0}-1}\right) + 1 \tag{A11}$$

$$\Omega_Q(S_T) = B(1,\beta) \quad \Leftrightarrow \quad P_Q(T) = 1 - ((\beta-1)QT/S_0 + 1)^{\frac{\beta}{1-\beta}} \tag{A12}$$

where $W(\cdot)$ is the Lambert-W function.

## A2  Accounting for discretization effects

To obtain a discrete form of the analytical solution we can make two assumptions. First, that the timeseries of inflow concentrations, and of water inflows and outflows are in fact constant within a timestep $\Delta t$, so

$$\mathtt{C_{Jj}} = C_J(t) \text{ for } j \leq t/\Delta t < j+1 \tag{A13}$$

$$\mathtt{J}_j = J(t) \text{ for } j \leq t/\Delta t < j+1 \tag{A14}$$

$$\mathtt{Q}_j = Q(t) \text{ for } j \leq t/\Delta t < j+1 \tag{A15}$$

$$\tag{A16}$$

where $\mathtt{C_{Jj}}$ and $\mathtt{C_{Qj}}$ are the discrete forms of $C_J(t)$ and $C_Q(t)$.

Second, we assume the numerical estimates of the outflow concentration timeseries should reflect the average value of the analytical solution over each timestep. That is:

$$\mathtt{C_{Qj}} = \frac{1}{\Delta t} \int\limits_0^{\Delta t} C_Q(j\Delta t + \nu) \, d\nu \tag{A17}$$

In the continuous form, $C_Q$ is obtained by the convolution of $p_Q$ with $C_J$, as shown in equation (1). If we assume that the input concentrations are constant over each timestep then (1) can be expressed as the sum of integrals over each timestep



interval $j\Delta t \leq t < (j+1)\Delta t$, plus the 'old water' contribution:

$$
\begin{aligned}
C_Q(t) &= \int_0^t C_J(t-T)p_Q(T,t)\,dT + C_{old}P_Q(t,t)\\
&= \left( \sum_{i=0}^{j-1} \int_{t-(j-i))\Delta t}^{t-(j-i-1)\Delta t} C_J(t-T)p_Q(T,t)\,dT \right) + \int_0^{t-j\Delta t} C_J(t-T)p_Q(T,t)\,dT + C_{old}P_Q(t,t)\\
&= \left( \sum_{i=0}^{j-1} \mathtt{C}_{\mathtt{J}j-i-1} \int_{t-(j-i))\Delta t}^{t-(j-i-1)\Delta t} p_Q(T,t)\,dT \right) + \mathtt{C}_{\mathtt{J}j} \int_0^{t-j\Delta t} p_Q(T,t)\,dT + C_{old}P_Q(t,t)\\
&= \left( \sum_{i=0}^{j-1} \mathtt{C}_{\mathtt{J}j-i-1} \left( P_Q(t-(j-i-1)\Delta t,t) - P_Q(t-(j-i)\Delta t,t) \right) \right) + \mathtt{C}_{\mathtt{J}j} P_Q(t-j\Delta t,t) + C_{old}P_Q(t,t)
\end{aligned}
\tag{A18}
$$

To obtain the discrete outflow concentrations we must apply the timestep-averaging in (A17) to (A18), which yields the discrete convolution:

$$
\mathtt{C}_{\mathtt{Q}j} = \sum_{i=0}^{j} \mathtt{C}_{\mathtt{J}j-i}\,\mathtt{p}_{\mathtt{Q}i,j}\Delta t + C_{old}\mathtt{P}_{\mathtt{Q}j,j}
\tag{A19}
$$

where the timestep-averaged TTD $\mathtt{P}_{\mathtt{Q}i,j}$ is given by:

$$
\mathtt{P}_{\mathtt{Q}i,j} = \frac{1}{\Delta t} \int_0^{\Delta t} P_Q(i\Delta t+\nu, j\Delta t+\nu)\,d\nu
\tag{A20}
$$

and $\mathtt{p}_{\mathtt{Q}i,j}$ is obtained via the discrete derivative:

$$
\mathtt{p}_{\mathtt{Q}i,j} =
\begin{cases}
\dfrac{\mathtt{P}_{\mathtt{Q}i,j}}{\Delta t} & i = 0\\[3mm]
\dfrac{\mathtt{P}_{\mathtt{Q}i,j} - \mathtt{P}_{\mathtt{Q}i-1,j}}{\Delta t} & i > 0
\end{cases}
\tag{A21}
$$

For the elementary case of steady flow and uniform sampling this gives:

$$
\mathtt{p}_{\mathtt{Q}i,j} = \frac{1}{\Delta t\delta} \times
\begin{cases}
\delta + e^{-\delta} - 1 & i = 0\\[2mm]
e^{-(1+i)\delta}\left(e^\delta - 1\right)^2 & i \geq 1
\end{cases}
\tag{A22}
$$

where $\delta = \Delta t Q/S_0$. Other forms are given in Table 4. Note that in the Exponential an Biased Young cases $\delta$ must be less than 1.

*Author contributions.* CJH contributed to the conceptualization, methodology, formal analysis, investigation, software development, validation/evaluation, visualization, original draft preparation, funding acquisition, project administration, and supervision. EXF contributed to the software development, validation/evaluation, visualization, original draft preparation, review  editing





*Competing interests.* The contact author has declared that none of the authors has any competing interests.

*Disclaimer.* Publisher's note: Copernicus Publications remains neutral with regard to jurisdictional claims in published maps and institutional
affiliations.

*Acknowledgements.* Thanks to Oliver Evans and Fei Lu for their contributions to the mesas.py code. This work was supported by a U.S. National Science Foundation grant (EAR-1654194)



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
