# Peer review of "mesas.py v1.0: A flexible Python package for modeling solute transport and transit times using StorAge Selection functions"

_EGUsphere, 2022_

## Author Comment (AC1)

We would like to thank the reviewers for their comments, which have improved the manuscript. Below we have responded to the comments provided. It is unfortunate that both had difficulties installing the software. We agree that it is essential that they be able to in order to provide a review of the manuscript. Hopefully there are not the same difficulties now. If there are, please contact us and we will address them immediately so that the review can continue.

**Reviewer 1**

*The paper by Harman and Xu Fei is a great contribution to the field of hydrology and solute transport. The software they describe in the manuscript is well packaged, fully-documented and it is flexible to several possible user needs. The code is shown to provide accurate numerical solutions against meaningful benchmarks. The paper clearly illustrates the model capabilities and provides the readers with novel benchmark analytical and discretized solutions. I only have minor suggestions for improvement; therefore, I am glad to recommend the paper for publication on GMD after minor revisions.*

Thank you very much for this assessment!

*Unfortunately, I was not able to install the latest version on any windows machine. I tried many times to conda-install mesas on a fresh, base Anaconda environment, but I got environment inconsistency problems. Conda automatically iterated over previous mesas version and the first version that it was able to install was version 0.2021.0909. It would be great if this issue could be checked before the paper is accepted.*

This is unfortunate! We were able to able to install using the following commands:

```
conda create -n mesas-env python=3 -y
conda activate mesas-env
conda config --append channels conda-forge
conda install mesas -y
```

*The authors stress in the abstract that the mesas implementation "provides a 15x reduction in mass balance errors compared to a previous implementation of SAS" (i.e. the tran-SAS implementation). While this is true, it also seems to be an unbalanced selection of the results, since for other metrics and parameters the difference is not always as large. Figure 3e shows that the difference between*

*the two implementation is the largest for k=1, but there are also values of k for which the two implementations have identical performances.*

We have tempered this language in the abstract to be more balanced. It should also be noted that it is no longer the case that there are values of k for which the two implementations have identical accuracy. In the process of revising the manuscript we discovered some small bugs that were introducing error into our method. After correcting these the accuracy of the method increased for the cases where it was previously similar to tran-SAS.

*There is no doubt that mesas is generally more accurate than the Euler-Forward-based implementation of tran-SAS, but I think it would be fair to: 1) compare the computational times in addition to the numerical accuracy; 2) if possible, make a comparison with the higher-order implementation of tran-SAS.*

Thank you for these suggestions, which we have implemented in the revised manuscript. Section 4.3 compares the perfomance of mesas.py and tran-SAS, with the latter usually faster for longer timeseries, and decidedly so for timeseries longer than about 1000 timesteps. Performance is similar for short timeseries.

We have also included the higher-order implementation of tran-SAS in the comparison against the benchmark in Figure 4. It's performance is similar to the regular tran-SAS. This is because in both cases tran-SAS reports the instantaneous TTD at the end of the timestep. The benchmark used is the timestep-averaged TTD, which is essential for accurate mass accounting. The mesas code is designed to estimate the timestep-averaged TTD, and thus can achieve higher performance against this benchmark. When we look at the instantaneous TTD predicted by we see that the distribution of errors is similar to that of tran-SAS (Figure 4b,d).

- 29: "sophisticated", I am not sure how to interpret this word in this context

  Neither am I! Removed.

- 46: here it is mentioned that "solute/tracer storage and outflow rates as part of the solution, not through a subsequent convolution integral", but I find this sentence possibly inaccurate. Solute storage rates do not need a convolution integral while for solute outflow rates it seems to me that equation (13) is in fact a convolution-like equation.

  I think I disagree with the reviewer here, though I do see where they are coming from. If the concentration term $C_T(T, t)$ (see equation 9) that goes into $\dot{m}_q(T, t)$ in equation (13) were replaced with the input concentration $C_J(t - T)$ then this would indeed be a convolution, though one where the kernel of the convolution $p_Q(T, t)$ changes with time. While $C_T(0, t) = C_J(t)$, fractionation when $\alpha(t) \neq 1$ can cause it to change over

time. With both terms time-variable it seems like a stretch to call it a convolution.

Nevertheless, rather than clarifying this subtlety I have reworded the sentence to simply "*uses a novel mass-tracking approach that estimates solute/tracer mass storage as part of the solution*".

- 114: In some circumstances, small quantities of chloride can be taken up by plants, so I would not assume the concentration in the evapoconcentration "must" necessarily be 0. See Xu, G., H. Magen, J. Tarchitzky, and U. Kafkafi (1999), Advances in chloride nutrition of plants,Adv. Agron.,68, 97–150, doi:10.1016/S0065-2113(08)60844-5
  It may be taken up through the roots of plants, but there in the plants it remains. It is not lost through the stomata as evapotranspiration.

- 150—156 (Eq 15—17): I don't get the notation using the delta symbol $\delta$ instead of the traditional d. Why can one not jump from Eq. (15) to Eq. (17) simply by diving by dT?
  Agreed. I'm not sure what the point of this was. Omitted.

- 182: "The accuracy of the results obtained this way may be poor". Can you expand on why? A user would want to understand this.
  Thank you, we have added "*The piecewise linear form may be inaccurate where the pdf changes rapidly (as a Gamma distribution does near $S_T = 0$ when $\alpha < 1$), or where it approaches an asymptotic value at the tails.*"

- 183: I'd suggest to remove "Uniform" from the section's title because one may be induced to think that there is a uniform CDF available in the code
  It has been removed from the title and a sentence has been added: "*When $N = 1$ this is simply a uniform distribution.*"

- Table 4 is very useful
  Thank you!

- 368-378: In figure 1r the error seems to be nonstationary and fast-growing. Perhaps a longer simulation test is needed in this case to quantify the correct error magnitude.
  This comment revealed a small issue with out code, and a small error in the partial piston analytical solution, which has been corrected. The overall errors do not grow, but rather stabilize.

- 398 (Figure 2): I find it a bit confusing to present in the same figure a benchmark comparison and a demonstration of the model solution for different parameters. I recommend to separate these subplots into different figures.
  Done

- 40: implementation
  Fixed

- 103: for clarity, consider spelling out C_Q instead of starting the sentence with "it"
  Done

- 141: whose with
  Fixed

- 171: Sm
  Fixed

- Table 2 and 3: these different tables have the same caption: is this on purporse? If yes, perhaps it will be good to rename them to 2a and 2b upon article production
  This was an error. Fixed.

- Figure 3: There are no subplot indices a), b) etc. . . in the figure.
  Fixed.

- Figure 3 caption: "an high-accuracy"
  Fixed

**Reviewer 2**

*The manuscript provides a comprehensive walkthrough of mesas, a program that utilizes the StorAge Selection (SAS) function to simulate time-varying transport dynamics. It is well-written and strikes a good balance between technical specifics and instructions for users.*

*Despite the manuscript being well-written, I encountered difficulties in reviewing it due to the problem in installing the program, as stated in comment #1. As a reviewer for the GMD journal, I understand the importance of testing the program and hence, I must defer making any recommendations until I can successfully install and run the program. Additionally, I have provided some suggestions for improvement in comment #2.*

*In L50-51 on page 2, the authors mentioned that the program could be installed through conda, with the code available on conda-forge. However, despite following the instructions and attempting to install the mesas package through "conda install -c conda-forge mesas", an error occurred. The error message read as follows: "PackagesNotFoundError: The following packages are not available from current channels: - mesas".*

This is confusing, since the `-c conda-forge` option should add the conda-forge channel. Try running `conda config --append channels conda-forge` first.

*Despite my attempts to manually install the program by following the instructions provided on the GitHub channel (L442), the installation process failed. While I was able to successfully compile the Fortran code, I encountered issues while attempting to install mesas using either "Python setup.py install" or pip. In the former case, I received "7 warnings and 9 errors generated" while in the latter, the error message read "ERROR: Failed building editable for mesas. Failed to build mesas. ERROR: Could not build wheels for mesas, which is required to install pyproject.toml-based projects".*

*In addition, I attempted to install the program using the source code available on Zenodo (L442). The absence of CMakeLists.txt in cdflib90 prevented me from compiling the Fortran code. Running "python setup.py install" resulted in the error message "Fatal Error: Cannot open module file 'cdf_gamma_mod.mod' for reading at (1): No such file or directory".*

Unfortunately the instructions on the github site are out of date. This has been corrected. Up to date instructions for building from source are in the documentation at https://mesas.readthedocs.io/en/latest/installation.html.

- Potential issues with the pre-determined shape of the SAS functions: If I understand it correctly, the shape of the SAS function must be determined a priori in mesas before simulating the dynamics. There are some issues with such a priori determination (e.g., Harman, 2019). I think mentioning the issues would be beneficial for potential users.
  Added "The SAS functions needed to represent a particular system are typically obtained by first choosing a functional form from those presented below, and then tuning the parameters of that functional form such that the model predictions match the tracer observations. It should be noted that there is currently an element of subjectivity and imprecision here, as multiple functional forms may produce equally acceptable fits to the available data (Harman 2019). In previous applications to watersheds, streamflow has been represented with a heavily right-skewed distribution whose mean varies inversely with catchment wetness, and ET by uniform distributions over the youngest water in storage. More physically-based parameterizations may be available in the future. "

- The required version of python must be specified.
  It is now specified in the code availability statement

- L77: a outflow → an outflow
  Fixed

- L111-L128: It would be beneficial if the authors could offer guidance on how to simulate isotopic fractionation using alpha_q. Although L115-118 alludes to the possibility of accounting for isotopic fractionation using alpha_q, it is unclear how to define the variable to accomplish this.
  Thanks. This has been added: "Note that if isotope fractionation is being modeled, the isotope data must be given as an isotope ratio, rather than a

$\delta$ value (per mille). For example, values of $\delta^{18}$O must be converted using $R = \delta^{18}$O$/1000 + 1$. The value of $\alpha$ is simply the regular fractionation factor $\alpha_{A/B}$ (Kendall and Campbell 1998). If fractionation is being neglected $\delta$ values may be used."

- L131: (11) $\rightarrow$ (11).
  Fixed

- L132-135: Providing descriptions for the practical necessity of T_max would be useful.
  "$T_{max} = t$ usually, but may be set to be less than $t$ to reduce memory demands or speed up computation (at the cost of potentially truncating the contributions of water that entered early in the simulation to outflows later)."

- L137: Would be easier to read if the number of equations and the number of unknowns were provided.
  This has been changed to "The equations above cannot be solved on their own, as $p_Q$ is not known. The SAS functions provide the required additional relationship linking the age-ranked storage and the transit time distribution."

- L208: Please define S_i(t).
  "For notational consistency we can define the cumulative version of this as $S_i(t)$, but it is precisely equal to $S_T(t - t_i, t)$"

- Section 3.2.1: The example on page 12 uses two pdfs for the discharge SAS function, but it wasn't clear how the weight of each pdf was considered.
  This is discussed in the second paragraph below the example: "Each of the keys naming a bulk flux in `sas_specs` is associated with a dictionary specifying the SAS functions for that flux. That dictionary can also include multiple SAS functions, which are combined together using time-varying weights. In the example above would expect to find columns in the timeseries dataset titled `"Q SAS function 1"` and `"Q SAS function 2"` containing weights to multiply each SAS function. These weights should add up to 1, though this is not checked. If the dictionary associated with each flux contains only one key:value pair then it is not necessary to provide a weights column in the timeseries dataset."